# Measurement of Skin Thickness Using Ultrasonography to Test the Usefulness of Elastic Compression Stockings for Leg Edema in Pregnant Women

**DOI:** 10.3390/healthcare10091754

**Published:** 2022-09-13

**Authors:** Airi Banba, Masafumi Koshiyama, Yumiko Watanabe, Koji Makino, Eri Ikuta, Nami Yanagisawa, Ayumi Ono, Miwa Nakagawa, Keiko Seki, Shin-ichi Sakamoto, Yoko Hara, Akira Nakajima

**Affiliations:** 1Graduate School of Human Nursing, The University of Shiga Prefecture, Hikone 522-8533, Japan; 2Department of Women’s Health, Graduate School of Human Nursing, The University of Shiga Prefecture, Hikone 522-8533, Japan; 3School of Nursing, Tsuruga Nursing University, Fukui 914-0814, Japan; 4School of Engineering, Department of Electronic Systems Engineering, The University of Shiga Prefecture, Hikone 522-8533, Japan; 5Jinno Ladies Clinic-Branch Hospital “Alice”, Hikone 522-0057, Japan

**Keywords:** skin thickness, ultrasonography, leg edema, pregnancy, elastic compression stockings

## Abstract

Background: One of the most common treatments for leg edema during pregnancy is the use of compression stockings. The purpose of this study was to evaluate the objective effectiveness in pregnant women, by measuring the changes of skin thickness using ultrasonography. Methods: Pregnant women were diagnosed with leg edema using the pitting edema method at 36 weeks of gestation. Twenty-four pregnant women (48 legs) with leg edema spent time without wearing elastic stockings at 36–37 weeks of gestation. Then, they wore elastic stockings for one week at 37–38 weeks of gestation. We measured the grade of edema (from 0 to 3) and the skin thickness of the lower leg by portable ultrasonography at 36, 37, and 38 weeks of gestation (a before-and-after study). Results: In 24 pregnant women, thigh edema was not detected in any of the 48 legs before or after the use of elastic stockings. All 48 legs in 24 pregnant women had physiological lower leg edema, but not thigh edema. The average grade of pitting edema in each lower leg significantly decreased after using the stockings (36 weeks, 1.77 ± 0.85; 37 weeks, 1.79 ± 0.77; 38 weeks, 1.04 ± 0.74, *p* < 0.0001). In addition, the skin thickness of the lower legs was significantly decreased after the use of elastic stockings (36 weeks, 7.47 ± 2.45 mm; 37 weeks, 7.93 ± 2.83 mm; 38 weeks, 7.15 ± 2.35 mm, *p* < 0.0001). Conclusions: The wearing of elastic compression stockings on the lower legs is objectively effective for improving leg edema in pregnant women.

## 1. Introduction

Edema is defined as the accumulation of fluid in the intercellular tissue that results from an abnormal expansion in interstitial volume [1]. The accumulation of skin fluid occurs when local or systemic conditions disrupt equilibrium, following increased capillary hydrostatic pressure, increased plasma volume, decreased plasma oncotic pressure (hypoalbuminemia), and increased capillary permeability [2].

During normal pregnancy, complete body water increases by 6–8 L, 4–6 L of which is extracellular, of which no less than 2–3 L is interstitial [3]. Thus, the most basic reason for leg edema in pregnancy is physiological edema. Leg edema that is not associated with pre-eclampsia is found in approximately 80% of all pregnancies [4]. Hormonal changes, including increased levels of progesterone, estrogen, and prolactin, cause changes in vascular permeability, promoting edema [3]. Furthermore, the enlarged uterus presses the inferior vena cava. Thus, an increased plasma volume and increased capillary permeability are caused by local venous hypertension from compression in pregnant women [2,5]. For these reasons, women in the late phase of pregnancy are likely to develop leg edema.

If leg edema is left untreated, it can cause increasingly painful swelling, difficulty walking, stiffness, stretched skin (which can become itchy and uncomfortable), increased risk of infection, decreased blood circulation (including blood concentration and thrombus), increased risk of skin ulcers, and other conditions [6].

Treatments used for leg edema include leg elevation, water immersion, massage, intermittent pneumatic compression, use of medication, reflexology, bandage, and elastic stockings [7,8]. Of these treatments, the most common treatments for leg edema during pregnancy are leg elevation and compression stockings [9]. However, there are few reports regarding them [10,11]. The objective of this study was to evaluate the effectiveness of elastic compression stockings for leg edema in pregnant women, measuring the changes of skin thickness using ultrasonography. 

There are quantitative techniques that may also be used to assess leg edema, including the measurement of leg circumference or ankle/calf diameter [12,13]. However, these methods assess leg edema lack sensitivity and are of most value in the assessment of relative changes in leg edema. In brief, these measurements contain operator errors.

We used portable ultrasonography to evaluate the effectiveness of elastic compression stockings for leg edema in pregnant women, because ultrasonography is a reliable, objective, and quantitative method for identifying legs with and without edema and is without operator variability [14]. The ultrasound device is portable, the imaging method is easy to use and produces consistent results between operators. 

## 2. Materials and Methods

### 2.1. Study Subjects and Study Approval

Twenty-four pregnant women (primiparas, n = 15, multiparas, n = 9) who attended an outpatient clinic in the department of Obstetrics and Gynecology, Hikone City, Shiga, Japan at 36 weeks of gestation and who were clinically presented with lower leg edema (the anterior surface of the tibia) were selected between June 2021 and September 2021. The study design was a before-and-after study. The study protocol was approved by the Ethics Committee of the University of Shiga Prefecture (no. 811), and all pregnant women gave their written informed consent prior to study entry. 

### 2.2. Clinical Characteristics of the Study Participants

The mean age of the pregnant women ± standard error of the mean (SEM) was 31.9 ± 5.9 years (range, 22–49 years). The body mass index (BMI) prior to pregnancy was calculated from the pre-pregnancy height (in m) and weight (in kg). Thus, the mean BMI of the overall population was 21.3 ± 2.9 kg/m^2^. 

The subjects of this study were 24 pregnant women with bilateral lower leg edema at 36 weeks of gestation.

### 2.3. Assessment and Grading of Edema of the Leg before and after Wearing Elastic Compression Stockings

Pregnant women were diagnosed with lower leg edema using the pitting edema method at 36 weeks of gestation. Finger pressure was applied to the swollen area of the lower leg skin and the thigh (two points) to determine whether an indentation formed that persisted after the removal of pressure. This method was used at 36, 37, and 38 weeks to determine whether or not skin edema was present. All selected pregnant women with lower leg edema spent time without wearing elastic compression stockings on the leg at 36–37 weeks of gestation. Then, the women wore elastic compression stockings for one week at 37–38 weeks of gestation. We asked the participants not to alter their lifestyle during the study period.

We used the pitting edema method to determine the grade of skin edema on the thigh and lower leg (the anterior surface of tibia) and measured changes of the lower leg skin thickness comparing before and after wearing elastic compression stockings (Table 1). In brief, we used these methods three times at 36, 37, and 38 weeks of gestation.

The grades of pitting edema were as follows: Grade 0, negative for edema, with no persisting indentation after release of finger pressure; Grade 1, mild pitting edema (slight indentation) that disappeared within 10 s; Grade 2, moderate pitting edema that disappeared after 10–15 s; Grade 3, severe pitting edema that lasted for more than 15 s, after the release of finger pressure (Table 2) [14].

### 2.4. Measurements of the Skin Thickness in Leg Edema before and after Wearing Elastic Compression Stockings

The thickness of the skin of the leg was measured using a B-scan portable ultrasound device (Viamo sv7, Canon Medical Systems Co., Tochigi, Japan) (Figure 1A). The skin thickness measurements included the epidermis, dermis, and subcutaneous tissue above the fascia and the muscle of both legs [14]. We used a convex probe in reference [12]. In this study, we changed the probe to a linear type. This probe was suitable to measure the skin. The same trained evaluator (who had been trained for three months) assessed the measured values under the appropriate conditions. After sitting and stretching women’s legs, the measurements were done in a room at a certain temperature. The study was conducted in the morning.

Ultrasonography was performed three times in the same women at 36, 37, and 38 weeks of gestation. A 10 MHz ultrasound linear probe was placed on the skin of the lower leg at a definite point, at 6 cm from the upper part of the medial malleolus and 1 cm inside the anterior border of the tibia (Figure 1B [14]). At this definite point, there were few other structures between the skin and bone, with a thin portion of soleus muscle. When the skin thickness was being measured, the women repeatedly performed anteflexion and retroflexion of the ankle [14]. Because the skin thickness measurements included the epidermis, dermis, and subcutaneous layers above the fascia and muscle, movements of the ankle resulted in the movement of the fascia and muscle layers, but not the skin, which allowed identification of the skin layer. Using ultrasonography, the distance between the skin surface and the upper part of the fascia was measured as the skin thickness, which included the epidermis, dermis, and subcutaneous tissue layers. We compared it between the same side leg before and after wearing stocking in each person (before–after study).

We also measured changes of the skin thickness before and after wearing elastic stockings (at 36, 37, and 38 weeks of gestation, Table 1). A pregnant woman could choose the stockings that fit her size (size S, M, or L) (Figure 1C). These elastic compression stockings (Akiyama Co., Tokyo, Japan) have medical applications and were all worn on the lower legs. The compressive pressure of a below-knee graduated elastic compression stockings was 18–27 mmHg (27 mmHg on the ankle and 18 mmHg on the calf). In addition, we provided a questionnaire that asked “What do you think about the merits and demerits of wearing compression stockings?”

### 2.5. Statical Analyses

The statistical analyses were performed using the JMP Pro statistical software program, version 14 (SAS Institute Japan, Tokyo, Japan) and GraphPad Prism, version 9 (GraphPad Software, San Diego, CA, USA). The results of skin thickness are presented as the mean ± SEM (mm) and were compared using a one-way repeated measures ANOVA and paired *t*-test before and after wearing elastic compression stockings. The changes in the grade of edema were calculated using the chi-squared test, repeated measures Kruskal–Wallis test and paired Wilcoxon test. The data from one week before wearing stockings were considered the control values. We then compared these data with the data obtained after the intervention (before-and-after study). We calculated the effect size of the study (minimum sample size) using an α level of 5% and power of 90%. Thus, the required sample size was calculated as 10 women (20 legs) based on an analysis of the mean ± SEM measurements of the thickness of the skin of the 10 legs in 5 women with edema before wearing stockings vs. 10 legs in the same 5 women after wearing stockings (α level, 5%; power, 90%). *p*-values of <0.05 were considered to indicate statistical significance.

## 3. Results

### 3.1. Level Changes in the Grade of Pitting Edema from before to after the Wearing of Elastic Compression Stockings

All 24 pregnant women with 48 legs had physiological leg edema without pre-eclampsia (Table 3). In addition, thigh edema was not detected in any of the 48 legs or after the wearing of elastic compression stockings. On the other hand, pitting edema disappeared in 10 of 48 (21%) lower legs after wearing stockings. In addition, the grade of pitting edema significantly decreased (Table 3, *p* < 0.0001).

In the lower legs, the average grade of pitting edema in each leg significantly decreased after wearing the stockings (36 weeks: 1.77 ± 0.85 vs. 37 weeks: 1.79 ± 0.77 vs. 38 weeks: 1.04 ± 0.74, *p* < 0.0001). In addition, the changes in the grade of lower leg edema also decreased (before wearing stockings: 0.02 ± 0.39 vs. after wearing stockings: −0.75 ± 0.79, *p* < 0.0001, Figure 2).

### 3.2. Level Changes of the Lower Leg Skin Thickness from before to after the Wearing of Elastic Compression Stockings

Next, we measured the skin thickness of lower legs before and after wearing elastic compression stockings. We present typical skin photographs of the same right leg in the same woman at 36, 37, and 38 weeks of gestation (Figure 3). The thickness of the right leg skin changed from 9.4 mm, after reaching 10.7 mm (just before wearing stockings), and decreased to 7.4 mm (just after wearing stockings). We compared the individual changes of skin thickness using one-way repeated measures ANOVA. These levels at 36, 37, and 38 weeks of gestation were 7.47 ± 2.45 mm, 7.93 ± 2.83 mm, 7.15 ± 2.35 mm, respectively, which was statistically significant (*p* < 0.0001) (Figure 4). In addition, level changes of the lower leg skin thickness from before to after the wearing of elastic compression stockings significantly decreased by a paired *t*-test (0.34 ± 0.71 mm vs. −0.78 ± 0.99 mm, *p* < 0.0001) (Figure 5).

### 3.3. Women’s Impressions after Wearing Elastic Compression Stocking

We provided questionnaires of the merits and demerits to 24 participants with leg edema after intervention. Regarding the merits, 19 (79%) women said that their legs became very comfortable. Regarding the demerits, 11 (46%) women said that it was slightly difficult to wear the stockings and 4 (17%) women said that it was slightly hard to take off the stockings. This was due to the enlargement of their abdomens due to pregnancy.

## 4. Discussion

In the present study, we knew that all 48 legs in 24 pregnant women had physiological lower leg edema, but not thigh edema. Physiological leg edema is not likely to extend to the thigh, except in cases of pre-eclampsia. The study conducted by Suehiro et al. supports these data [15]. As the clinical severity of dependent edema progressed, the echogenicity increased in all parts of the lower extremities. As a result, it was sufficient to observe the effect of elastic compression stockings worn below the knee. It was also easier for a woman with an enlarged uterus during late pregnancy to wear stockings below the knee. From that point alone, we could expect a sufficient effect.

Physiological leg edema results from hormone-induced sodium retention. Edema may also occur when the enlarged uterus intermittently compresses the inferior vena cava in the supine position, obstructing outflow from both femoral veins [2,3,5]. Therefore, leg edema during late pregnancy can continue until the conclusion of the pregnancy. In the present study, the leg edema in pregnant women worsened from 36 to 37 weeks of gestation (pitting edema grade increased by 0.02 ± 0.39; skin thickness increased by 0.34 ± 0.71 cm). If leg edema rapidly worsens, the blood in the veins can become concentrated and may easily form blood clots. Therefore, we think that early treatment of leg edema in pregnant women is necessary to avoid these blood conditions and discomfort.

We used the pitting edema method and the measurement of skin thickness by ultrasonography as quantitative methods to evaluate the degree of edema. We previously reported that ultrasonography was a reliable, objective, and quantitative method for identifying legs with and without edema that was not affected by operator variability [14]. However, we faced a problem in that fatty legs showed thick skin, even in cases without edema [14,16]. Thus, this method is useful as quantitative judgment of the effects of treatment for women who are diagnosed with edema in advance. Prior to the use of ultrasound measurement, we should determine whether or not a woman has leg edema, using the pitting edema method.

We previously reported that there are two ways of achieving interstitial fluid movement in the treatment of subcutaneous edema of the legs [8]. One way involves the movement of fluid in the extravascular space of the legs. To move fluid this way, we can use leg elevation [17], foot massage [18], intermittent pneumatic compression [19,20], or reflexology [21,22]. Another way involves the movement of the fluid from the extravascular space into the venous system. To move fluid this way, we can use water immersion [23,24], medicine [25], bandages [26,27], or stockings [28,29]. Among these methods, we selected wearing stockings and investigated its effect using a quantitative measurement, because pregnant women could easily use them in everyday life and they were easily available due to their low-cost. 

In the present study, edema disappeared in 10 (21%) of 48 lower legs of pregnant women who wore elastic stockings. Whereas the grade of pitting method increased 0.02 ± 0.39 in treatment-free pregnant women with lower leg edema, it significantly decreased by −0.75 ± 0.79 in pregnant women wearing elastic stockings. Whereas the thickness of the lower leg skin increased by 0.34 ± 0.71 mm in treatment-free pregnant women, it significantly decreased by −0.78 ± 0.99 mm in pregnant women wearing elastic stockings. From these facts, it may be said that the wearing of elastic stockings on the lower legs was effective for improving edema in pregnant women. Elastic compression stockings can move subcutaneous fluid from the extravascular space into the venous system. Wearing compression stockings may also be effective for preventing lower leg edema.

From the results of the questionnaire survey of participants after wearing stockings, 79% of participants said that their legs became comfortable. Thus, even when the demerits were considered, wearing elastic compression stockings was useful for pregnant women with leg edema because it improved their quality of life, which had been reduced by leg edema. We hope that this study will help provide guidance on the early use of elastic stockings by pregnant women with edema. 

Several limitations associated with the present study warrant mention. This study was not a controlled before-and-after study. Because we could not establish another group that did not receive the treatment (wearing stockings) due to the ongoing COVID-19 pandemic, there was some risk of bias. We therefore set a period of one week before the subjects wore the stockings and compared the data from that period with the data from one week after wearing stockings (before-and-after study). In addition, we did not investigate the movement of fluid from the subcutaneous layer via an objective method. In the future, we should explore this point, as this is expected to reveal the true merits of wearing elastic compression stockings for leg edema in pregnant women. 

## 5. Conclusions

Physiological leg edema in pregnant women is not likely to extend to the thigh, except in cases of pre-eclampsia. It is sufficient to observe the effect of elastic compression stockings worn below the knee. The wearing of elastic compression stockings on the lower legs is objectively effective for improving lower leg edema in pregnant women. However, further investigations are needed to clarify the movement of fluid from the lower leg skin after treatment. 

## Figures and Tables

**Figure 1 healthcare-10-01754-f001:**
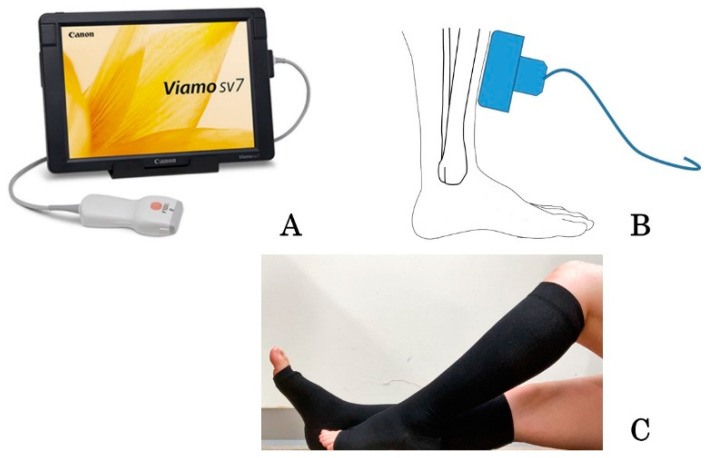
The portable ultrasound device, its use in measuring the skin thickness, and the wearing of compression stockings. (**A**) B-scan portable ultrasound device, Viamo sv7; (**B**) the lower edge of the upper part of the medial malleolus and 1 cm inside the anterior border of the tibia [14]; (**C**) the pregnant women with leg edema wore the elastic compression stockings below the knee.

**Figure 2 healthcare-10-01754-f002:**
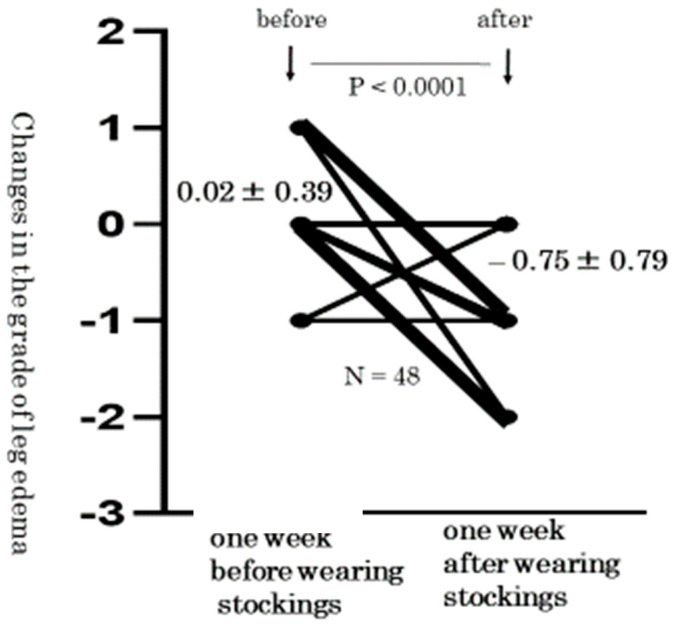
Changes in the grade of pitting edema. The grade of lower leg edema also decreased (before wearing stockings: 0.02 ± 0.39; after wearing stockings: −0.75 ± 0.79, *p* < 0.0001).

**Figure 3 healthcare-10-01754-f003:**
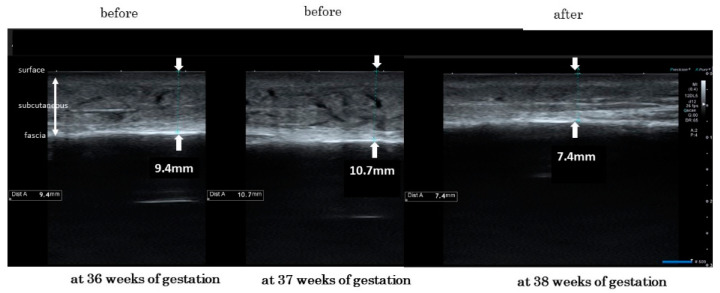
Typical skin changes in photographs of the same right leg in the same woman at 36, 37, and 38 weeks of gestation (before and after wearing stockings).

**Figure 4 healthcare-10-01754-f004:**
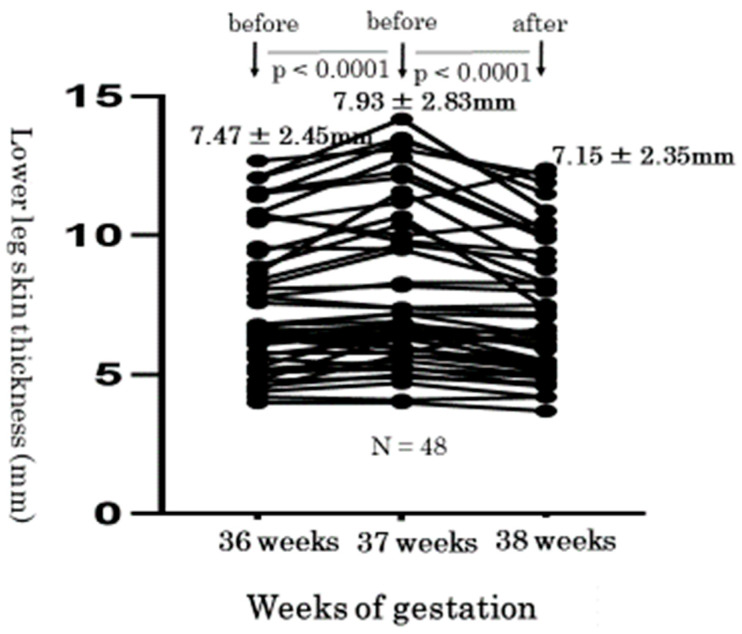
Transition of the lower leg skin thickness with edema at 36, 37, and 38 weeks of gestation. The comparison of the thickness at 36, 37, and 38 weeks of gestations revealed a statistically significant differences (36 weeks: 7.47 ± 2.45 mm; 37 weeks: 7.93 ± 2.83 mm; 38 weeks: 7.15 ± 2.35 mm, *p* < 0.0001).

**Figure 5 healthcare-10-01754-f005:**
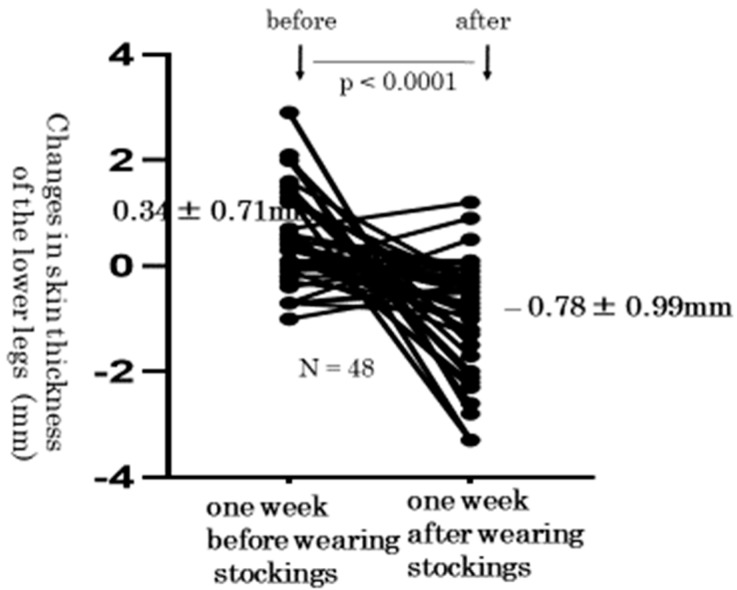
Changes in skin thickness of the lower legs. The lower leg skin thickness also showed a significant change from before to after the wearing of elastic compression stockings (0.34 ± 0.71 mm vs. −0.78 ± 0.99 mm, *p* < 0.0001).

**Table 1 healthcare-10-01754-t001:** Measurements of the skin thickness in leg edema and wearing elastic compression stockings.

	At 36 Weeks	At 37 Weeks	At 38 Weeks
pitting edema method	on the thigh and lower leg	on the thigh and lower leg	on the thigh and lower leg
measurments of the skin thickness	in lower leg	in lower leg	in lower leg
wearing elastic compression stockings		put on	pull off

**Table 2 healthcare-10-01754-t002:** The grades of pitting edema.

Grade of Skin Edema	0	1	2	3
status of pitting edema	negative for edema	pitting edema that disappeared within 10 s	pitting edema that disappeared after 10–15 s	pitting edema that lasted for more than 15 s

**Table 3 healthcare-10-01754-t003:** Level changes in the grade of pitting edema from before to after the wearing of elastic compression stockings.

	Thigh				Lower Leg			
grade of edema	0	1	2	3	0	1	2	3
no. of legs at 36 weeks gestation	48	0	0	0	0	24	11	13
no. of legs at 37 weeks gestation	48	0	0	0	0	20	18	10 *
no. of legs at 38 weeks gestation	48	0	0	0	10	28	8	2 *

Grade 0: negative for edema; Grade 1: mild pitting edema; Grade 2: moderate pitting edema; Grade 3: severe pitting edema. ^*^
*p* < 0.0001.

## Data Availability

Not applicable.

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
