# Peer review of "Measurement of Skin Thickness Using Ultrasonography to Test the Usefulness of Elastic Compression Stockings for Leg Edema in Pregnant Women"

_healthcare, 2022, doi:10.3390/healthcare10091754_

Round 1
Reviewer 1 Report (New Reviewer)
The authors are presenting their work on quantifying the effects of compression stockings on lower leg edema in pregnant women. This work is very similar to a previous work by Saliba-Junior et al. (see citation below) that also looked at the effect of compression stocking on leg edema in pregnant women. The only difference in this work is the use of ultrasound to measure skin thickness as a metric for edema, rather than diameter measurements, but the conclusions from both are effectively the same. While the skin thickness is a reasonable metric, the paper does not adequately describe why they chose this method over traditional objective measures of edema such as diameter (as used by Saliba-Junior et al.). In the absence of a clear justification for why their metric is better, there are no clear indications of the novelty of this work and it seems to repeat findings from previous studies with new techniques. Further, diameter measurements are the gold standard for edema measurement and sufficient justification is needed as to why the authors are selecting a potentially more complicated technique in its place. Apart from this major concern, I have a few other comments that need to be addressed, primarily related to the novelty of the work:
1. As mentioned before, the following paper needs to be cited since it is similar to the work communicated here. Significant work needs to be put into discussing how the described technique is better and to justify the novelty of the present work since the findings are the same:
Saliba-Júnior OA, Rollo HA, Saliba O, Sobreira ML. Positive perception and efficacy of compression stockings for prevention of lower limb edema in pregnant women. J Vasc Bras. 2022 Jan 31;21:e20210101. doi: 10.1590/1677-5449.210101. PMID: 35399341; PMCID: PMC8958436.
2. The main way this work differentiates itself from previous work is in measuring skin thickness after using elastic compression. As such, the title of the manuscript needs to be changed to reflect that. A general title, such as the one used, makes it sound similar to other work in the field and does not differentiate what is novel about this work.
3. Please put spaces between rows in the tables for ease of reading.
4. Please use appropriate references for where stock images were taken from, such as Figure 1.
5. Figure 2 is very confusing. I understand that a lot of the points representing the patients overlap, but it will be good if the authors can show the relative concentration of data points by changing the dot size, line width etc.
Author Response
We described the changed sentences with red color in Text.
Comments and Suggestions for Authors
The authors are presenting their work on quantifying the effects of compression stockings on lower leg edema in pregnant women. This work is very similar to a previous work by Saliba-Junior et al. (see citation below) that also looked at the effect of compression stocking on leg edema in pregnant women. The only difference in this work is the use of ultrasound to measure skin thickness as a metric for edema, rather than diameter measurements, but the conclusions from both are effectively the same. While the skin thickness is a reasonable metric, the paper does not adequately describe why they chose this method over traditional objective measures of edema such as diameter (as used by Saliba-Junior et al.). In the absence of a clear justification for why their metric is better, there are no clear indications of the novelty of this work and it seems to repeat findings from previous studies with new techniques. Further, diameter measurements are the gold standard for edema measurement and sufficient justification is needed as to why the authors are selecting a potentially more complicated technique in its place. Apart from this major concern, I have a few other comments that need to be addressed, primarily related to the novelty of the work:
<A> Thank you! They measured ankle and calf diameter using a tape. I don’t think that diameter measurements are the gold standard for edema measurement. It's doubtful whether or not they measure it accurately in millimeters. If a tape leans a little, the measurement contains errors in centimeter. Thus, we added the following section in Introduction:
There are quantitative techniques that may also be used to assess leg edema, including measurement of leg circumference or ankle/calf diameter [12, 13]. However, these methods for assessing leg edema lack sensitivity and are of most value in the assessment of relative changes in leg edema. In brief, these measurements contain operator errors.
We used portable ultrasonography to evaluate the effectiveness of elastic compression stockings for leg edema in pregnant women. Because ultrasonography is a reliable, objective, and quantitative method for identifying legs with and without edema and was without operator variability [14]. The ultrasound device is portable, the imaging method is easy to use and produces consistent results between operators.
- As mentioned before, the following paper needs to be cited since it is similar to the work communicated here. Significant work needs to be put into discussing how the described technique is better and to justify the novelty of the present work since the findings are the same:
Saliba-Júnior OA, Rollo HA, Saliba O, Sobreira ML. Positive perception and efficacy of compression stockings for prevention of lower limb edema in pregnant women. J Vasc Bras. 2022 Jan 31;21:e20210101. doi: 10.1590/1677-5449.210101. PMID: 35399341; PMCID: PMC8958436. <A> Thank you for comments. Yes, we added the following section:
There are quantitative techniques that may also be used to assess leg edema, including measurement of leg circumference or ankle/calf diameter [12, 13]. However, these methods for assessing leg edema lack sensitivity and are of most value in the assessment of relative changes in leg edema. In brief, these measurements contain operator errors.
We used portable ultrasonography to evaluate the effectiveness of elastic compression stockings for leg edema in pregnant women. Because ultrasonography is a reliable, objective, and quantitative method for identifying legs with and without edema and was without operator variability [14]. The ultrasound device is portable, the imaging method is easy to use and produces consistent results between operators.
- The main way this work differentiates itself from previous work is in measuring skin thickness after using elastic compression. As such, the title of the manuscript needs to be changed to reflect that. A general title, such as the one used, makes it sound similar to other work in the field and does not differentiate what is novel about this work. <A> Thank you for your good advice。We changed it to:
Usefulness of elastic compression stockings for leg edema in pregnant women, measuring the changes of skin thickness using ultrasonography
- Please put spaces between rows in the tables for ease of reading.
<A> Thank you for advice. Yes, we put spaces between rows.
- Please use appropriate references for where stock images were taken from, such as Figure 1. è <A> Thank you for comments. But, Figure 1 is our original one. The leg with stocking is first author’s one.
- Figure 2 is very confusing. I understand that a lot of the points representing the patients overlap, but it will be good if the authors can show the relative concentration of data points by changing the dot size, line width etc. è <A> Thank you for your advices. We deleted arrows.
Alternatively, We put thick lines.

Reviewer 2 Report (New Reviewer)
Congratulations to the authors for writing such interesting study on a topic with a paucity of information.
Few suggestions:
1) In Materials Methods: you mention 'diagnosed with lower leg edema' I suggest changing it to 'clinically presented with edema' as you are not really diagnosing with a pitting test.
2) it would have been interesting to know how many women it was their first or second or third pregnancy. I think this has influence on the physiological response.
3) Could you add / confirm that the women were measured at the same anatomical location at the same time of day? Time of day will be important. Was room temperature controlled? Did the women sit or lay down for certain amount of time before measurement commenced?
4) For how long did they wear the stockings before they reported significant decrease in edema?
Your discussion is strong and well written. You identify limitations of your study well and ideas for the future
Author Response
We described the changed sentences with red color in Text.
Comments and Suggestions for Authors
Congratulations to the authors for writing such interesting study on a topic with a paucity of information.
Few suggestions:
1) In Materials Methods: you mention 'diagnosed with lower leg edema' I suggest changing it to 'clinically presented with edema' as you are not really diagnosing with a pitting test.
<A> Yes, we changed it to clinically presented with lower leg edema
2) it would have been interesting to know how many women it was their first or second or third pregnancy. I think this has influence on the physiological response.
<A> Yes! We described “Twenty-four pregnant women (primiparas, n=15, multiparas, n=9)” in Material Methods. Primiparas mean first pregnancy, and multiparas mean over second pregnancy. There are no differences among them.
3) Could you add / confirm that the women were measured at the same anatomical location at the same time of day? Time of day will be important. Was room temperature controlled? Did the women sit or lay down for certain amount of time before measurement commenced?
<A> Thank you for good advices. We inserted the following sentences in Materials and Methods: After sitting and stretching women’s legs, the measurements were done in a room at a certain temperature. The study was done in the morning.
4) For how long did they wear the stockings before they reported significant decrease in edema?
<A> Yes! We described the following sentence in 2.3. Assessment and grading of edema of the leg before and after wearing elastic compression stockings.:
Then the women wore elastic compression stockings for one week at 37-38 weeks of gestation.
Your discussion is strong and well written. You identify limitations of your study well and ideas for the future <A> Thank you!!

Round 2
Reviewer 1 Report (New Reviewer)
Thank you for responding to the comments. I would like to still suggest one minor edit to the paper tile as “Measurement of skin thickness using ultrasonography to test the usefulness of elastic compression stockings for leg edema in pregnant women”. This will make it compact and get the main point across.
Author Response
<Referee 1、second>
Comments and Suggestions for Authors
Thank you for responding to the comments. I would like to still suggest one minor edit to the paper tile as “Measurement of skin thickness using ultrasonography to test the usefulness of elastic compression stockings for leg edema in pregnant women”. This will make it compact and get the main point across.
==><A> Thank you for good advice! Yes we changed our title to “Measurement of skin thickness using ultrasonography to test the usefulness of elastic compression stockings for leg edema in pregnant women”.

This manuscript is a resubmission of an earlier submission. The following is a list of the peer review reports and author responses from that submission.
Round 1
Reviewer 1 Report
Title
Title is appropriate because it is completely informative about the contents of the paper.
Abstract
The abstract respects the rules of the journal. The background and the aim are interesting. In the design is present the type of study. Methods need to be better explained. The clinical Impact is present but need to be better explained.
Text
The introduction and the discussion of the study clearly sum up the background of the study, but it can be improved by adding some reference. The authors provide a rationale for performing the study based on a review of the medical literature. Furthermore, they define well terms used in the remainder of the manuscript. The hypothesis is defined.
The methods are not clear.
What are the sample size and effect size of the study?
Who performed the ultrasound evaluation?
Better explain how the ultrasound was performed, with the landmarks used. Also explain better how the evaluation of the ultrasound images was performed.
What is the reliability of these measurements?
What are the questionaries regarding merits and demerits?
The study needs to be better explained in the methodology.
The number of references reported about the topic must be increased.
The results are reported clearly and concisely.
The discussion is similar to the introduction in the first part, it needs to be improved.
Introduce a paragraph for limitations.
The conclusions are not present, introduce a paragraph.
References
They are qualified and updated with the lasted data, but they must be integrated. The reference list follows the format for the journal.
Tables
They highlight the key points but they need to be improved.
Figures
They highlight the key points but they need to be improved.
Statistical Analysis
It isn’t needed further checking of data by a statistician reviewer.
General comments
The purpose of the study is original but the study needs to be improved in the introduction, methods and discussion. The hypothesis is defined. The methods are not clear. The study has been structured and carried out correctly but the methodology needs to be better explained. Discussion needs to be improved. Introduce a paragraph for limitations. The conclusions are not present, introduce a paragraph.
Author Response
We correct the text in red color.
Comments and Suggestions for Authors
Title
Title is appropriate because it is completely informative about the contents of the paper.
è(A) Thank you very much.
Abstract
The abstract respects the rules of the journal. The background and the aim are interesting. In the design is present the type of study. Methods need to be better explained. The clinical Impact is present but need to be better explained.
è(A) Thank you very much.
Text
The introduction and the discussion of the study clearly sum up the background of the study, but it can be improved by adding some reference. The authors provide a rationale for performing the study based on a review of the medical literature. Furthermore, they define well terms used in the remainder of the manuscript. The hypothesis is defined.
è(A) Thank you for your advices. We corrected the contents of text!
But, the topics of this methods are few.
The methods are not clear. What are the sample size and effect size of the study?
è(A) Thank you very much. We changed Table 1. And we added the following sentence in the text. We calculated effect size of the study (minimum sample size)using α level: 5% and power: 90%.
Who performed the ultrasound evaluation?
è(A) We add the following sentence in the text. A same measure evaluated the measured values under certain conditions.
Better explain how the ultrasound was performed, with the landmarks used. Also explain better how the evaluation of the ultrasound images was performed.
è(A) We explained the landmarks of measurement in the text. “A 10MHz ultrasound probe was placed on the skin of the lower leg at a definite point, including at 6cm from the upper part of the medial malleolus and 1cm inside the anterior border of the tibia (Figure 1B [12].” And we explained, “Using ultrasonography, the distance between the skin surface and the upper part of the fascia was measured as the skin thickness, which included the epidermis, dermis and subcutaneous tissue layer.” in Materials and Methods.
What is the reliability of these measurements?
è(A) We state the the reliability of measurements. “We previously reported that ultrasonography was a reliable, objective, and quantitative method for identifying legs with and without edema that was not affected by operator variability [12]. However, we faced a problem in that fatty legs showed thick skin, even in cases without edema [12, 13]. Thus, this method is useful as quantitative judgment of the effects of treatment for women who are diagnosed with edema in advance. Prior to the use of ultrasound measurement, we should determine whether or not a woman has leg edema, using pitting edema method.”in
Discussion.
What are the questionaries regarding merits and demerits?
è(A) We added : In addition, we provided questionnaires of “What do you think about the merits and demerits of wearing compression stockings?”in Materials and Methods.
The study needs to be better explained in the methodology.
è(A)We added some methodology in Materials and Methods.
We compared it between the same side leg before and after wearing stocking in each person. in Materials and Methods. And we changed Table 1.
The number of references reported about the topic must be increased.
è(A) Yes, we want to increase the topics of measurement of lower leg edema using ultrasonography. But we cannot find any paper about it. This study is very rare and valuable.
The results are reported clearly and concisely.
è(A) Thank you very much.
The discussion is similar to the introduction in the first part, it needs to be improve
è(A) Yes, we delete the following sentence: Physiological leg edema results from hormone-induced sodium retention.d.in Discussion.
Introduce a paragraph for limitations.
è(A) We added the following sentences in Discussion: This study has limitations. Because we did not investigate the movement of fluid from the subcutaneous layer with an objective method. In future, we should give light upon it. Then, the true merits of wearing elastic compression stockings for leg edema in pregnant women will be revealed.
The conclusions are not present, introduce a paragraph.
è(A) We added 5. Conclusion: Physiological leg edema in pregnant women is not likely to extend to the thigh, except in cases of pre-eclampsia. It is sufficient to observe the effect of elastic compression stockings worn below the knee.
The wearing of elastic compression stockings on the lower legs is objectively effective for improving leg edema in pregnant women. However, further investigations are needed to know the movement of fluid from the lower leg skin.
References
They are qualified and updated with the lasted data, but they must be integrated. The reference list follows the format for the journal.
è(A) Thank you very much.
Tables
They highlight the key points but they need to be improved.
è(A)Yes, we inserted arrows in Table 1.
Figures
They highlight the key points but they need to be improved.
è(A)Yes, we enlarged each figure and made it easier to see..
Statistical Analysis
It isn’t needed further checking of data by a statistician reviewer.
è(A) Thank you very much.

Reviewer 2 Report
Please see the word file.

Author Response
We correct the text in red color.
Summary . This original manuscript (article) addresses the issue regarding treatment of leg edema in the third trimester-pregnant women. The authors measured the thickness of the pretibial subcutaneous tissue on 10-MHz ultrasound before and after wearing the stockings, and claim the usefulness of the elastic stocking for leg edema in the pregnant women.
Evaluation I cannot easily accept the authors’ claim because the manuscript is suboptimal for a science article. My specific comments are described as follows.
- Study design a) There was no control group which is the most shortcoming of the article. Ideal control group consists of ipsilateral leg without wearing the elastic stocking. The authors used the data in the 48 legs in the 24 patients. How difference in the skin/subcutaneous thickness between the left and the right pretibial areas. If the difference is negligible, the data should be used the mean of right and left legs.
è(A) Thank you very much.for your comments. This is a paired two-group comparison test. Control was the left leg of human A before wearing stocking and intervention was the same-left leg of human A after wearing stocking. Control group consists of ipsilateral leg of the same woman. This design has not errors!! And we added the following sentence : We compared it between the same side leg before and after wearing stocking in each person. in Materials and Methods.
- b) I do not think that evaluation of pitting edema was need in the authors’ study.
è(A) Thank you very much for your kind comments. We checked the effect of wearing stocking just in case. We wanted to do double confirmation!!
- Ultrasound
- a) The authors describe ultrasound of “skin” for evaluation of edema throughout the text and captions of Fig.4. The subcutaneous tissue is not included in the skin. (epidermis/dermis). The authors should describe ultrasound of skin/subcutaneous tissue instead of ultrasound of skin.
è(A) No, no, no!! The white line of skin bottom is fascia!! Therefore, the distance between 2 arrows includes the epidermis, dermis and subcutaneous tissue. In the previous our-paper, we described the followings: When we measured the skin thickness, we had the women repeatedly perform anteflexion and retroflexion of the ankle. The skin consists of the epidermis, dermis and subcutaneous layers above the fascia and muscle. Therefore, when the women moved their ankle, the fascia and muscle layers moved as well, while the skin layer did not move allowing us to easily identify the skin layer (Yanagisawa, M.; Koshiyama,M.; Watanabe, Y.; Sato, S.; Sakamoto, S. A quantitative method to measure skin thickness in leg edema in pregnant women using B-scan portable ultrasonography: a comparison between obese and non-obese women. Med Sci Monit. 2019, 25, 1-9)
- b) In Materials and Methods section, the authors described “the distance between the skin surface and the upper part of the fascia”. In figure 4, the muscle beneath of the fascia (hyperechoic line) is not demonstrated. I guess the authors measured the distance between the skin surface to the tibial surface.
è(A) No, no, no!! As mentioned above, we checked the fascia, muscle and skin layer by moving woman’s ankle.
The white line on the bottom of skin is fascia!!
- Fig. 1-C.
I guess the figure is not demonstrated the edematous leg in a pregnant woman.
è(A)Thank you for comments. The echogenicity of edematous leg is not yet clear whether it is low or high. We now analyze echogenicity using ultrasound phot and photo-editing app. The only thing we can say is that the edematous skin increases its width.

Reviewer 3 Report
Abstract: I am confused by the abstract. line 17 says the women were diagnosed with leg edema and line 22 says thigh edema was not detected. Can you clarify this in your abstract. It may be clearer to just take out the line about thigh edema.
Although it is too late to change methods, how can you account for the change not being due to the subjects not walking as much at 38 weeks or some other variable. Would a better methodology have been to compare two groups at 37 weeks, one with stockings and one without. This would take away some of your covariates.
Can you please include an image of the ultrasound to demonstrate the measurement. Do the images in figure 1 need referencing or did you develop these yourself? Particularly B.
Did you perform any reliability testing. Was each method done only once or did you repeat and average?
The required sample size seems low to give a power of 90% is this worth double checking?
Line 136 - Should this say not detected in any of the 48 legs before or after.
Figures 2 and 3 are not at all clear and I don't feel add anything to the manuscript. I would suggest picking another way to demonstrate these results graphically.
The quality of the ultrasound images in Figure 4 is poor. It is difficult to see whether the deep fascia is actually the deep fascia or is this where your beam attenuated. I am unsure as to why it gets so dark below the skin fascia. Was the gain purposely turned down. When you are dealing with measurements in points of mm there can be substantial difference regarding where the cross bars of the measurement caliper is placed. Did you consider this and did you perform any reliability?
The font size of the wrodsFigure 4 Figure 5 and Figure 6 text are all different. Suggest making these the same. Figures 5 and 6 are very difficult to decipher.
The discussion states that reference 12 previously reported that ultrasonography was reliable, objective and quatitative and not afected by oeprator variability. I could not find evidence in reference 12 that supports this statement. Reference 12 also used a curved probe and not a linnear probe as you have appropriately used for this study.
Overall:
Well done. This is an interesting study. I have some concerns regarding the methodology. Firstly in terms of the methods themselves. Would it not have been beneficial to have a group to compare to that didn't have compression stockings at 37 weeks. I would suggest gathering data from such a group and comparing them. My second reservation is the lack of reliability testing for the measurements when dealing with such a small sample size. My final reservation is the presented ultrasound images which demonstrate no gained tissue beneath the fascial line. An independent person cannot make a judgement on whether this is the correct position of the fascial line.
The paper is written well and is beneficial in nature however the figures are not easily discernable and I would suggest finding a more appropriate way to display your results prior to publicaiton.
Author Response
We correct the text in red color.
Abstract: I am confused by the abstract. line 17 says the women were diagnosed with leg edema and line 22 says thigh edema was not detected. Can you clarify this in your abstract. It may be clearer to just take out the line about thigh edema.
è(A)Thank you for comments. We inserted the following sentence in Abstract. All 48 legs in 24 pregnant women had physiological lower leg edema, but not thigh edema.
Although it is too late to change methods, how can you account for the change not being due to the subjects not walking as much at 38 weeks or some other variable. Would a better methodology have been to compare two groups at 37 weeks, one with stockings and one without. This would take away some of your covariates.
è(A)Thank you for your good comments. This is a paired two-group comparison test. Control was the left leg of human A before wearing stocking and intervention was the same-left leg of human A after wearing stocking. Thus, we inserted the following sentence in Materials and Methods.
We asked the participants not to change their lifestyle during a study period.
Can you please include an image of the ultrasound to demonstrate the measurement. Do the images in figure 1 need referencing or did you develop these yourself? Particularly B.
è(A)Thank you for comments. Yes, this measurement was reported in previous our paper. So, we inserted its reference [12] in Figure 1B.
(B) The lower edge of the upper part of the medial malleolus and 1cm inside the anterior border of the tibia [12];
Did you perform any reliability testing. Was each method done only once or did you repeat and average?
è(A)Thank you for comments. This measurement was previously reported in 98 legs of pregnant women in our paper and tested repeatedly. [Reference 12].
The required sample size seems low to give a power of 90% is this worth double checking?
è(A) Yes, double checking was done. The data showed that significant differences were seen!!
Line 136 - Should this say not detected in any of the 48 legs before or after.
è(A) We can’t recognize line 136 (disappeared). The time from 36 weeks to 37 weeks was before and the time from 37 weeks to 38 weeks was after.
Figures 2 and 3 are not at all clear and I don't feel add anything to the manuscript. I would suggest picking another way to demonstrate these results graphically.
è(A) We are sorry that Figure 2 and 3 were not clear. Thus, we added all arrows on all bars.
The quality of the ultrasound images in Figure 4 is poor. It is difficult to see whether the deep fascia is actually the deep fascia or is this where your beam attenuated. I am unsure as to why it gets so dark below the skin fascia. Was the gain purposely turned down. When you are dealing with measurements in points of mm there can be substantial difference regarding where the cross bars of the measurement caliper is placed. Did you consider this and did you perform any reliability?
è(A) No, no, no!! The white line of skin bottom is fascia!! Therefore, the distance between 2 arrows includes the epidermis, dermis and subcutaneous tissue. In the previous our-paper, we described the followings: When we measured the skin thickness, we had the women repeatedly perform anteflexion and retroflexion of the ankle. The skin consists of the epidermis, dermis and subcutaneous layers above the fascia and muscle. Therefore, when the women moved their ankle, the fascia and muscle layers moved as well, while the skin layer did not move allowing us to easily identify the skin layer (Yanagisawa, M.; Koshiyama,M.; Watanabe, Y.; Sato, S.; Sakamoto, S. A quantitative method to measure skin thickness in leg edema in pregnant women using B-scan portable ultrasonography: a comparison between obese and non-obese women. Med Sci Monit. 2019, 25, 1-9)
In addition, there are cross bars of the measurements in 3 photos. It is hard to see them because they are very small!!
The font size of the wrodsFigure 4 Figure 5 and Figure 6 text are all different. Suggest making these the same. Figures 5 and 6 are very difficult to decipher.
è(A)Yes. Thank you for your good advices. We ask the journal office to arrange the Figure size.
The discussion states that reference 12 previously reported that ultrasonography was reliable, objective and quatitative and not afected by oeprator variability. I could not find evidence in reference 12 that supports this statement. Reference 12 also used a curved probe and not a linnear probe as you have appropriately used for this study.
è(A)Yes. Thank you for your advices. We used convex probe in reference 12. Exactly, this probe are not suitable to measure the surface skin. But errors (mm) of measurement are very few. In this study, we change the probe to linear type. This probe are suitable to measure the skin. In this study, we measured the skin thickness more accurately.
Overall:
Well done. This is an interesting study. I have some concerns regarding the methodology. Firstly in terms of the methods themselves. Would it not have been beneficial to have a group to compare to that didn't have compression stockings at 37 weeks. I would suggest gathering data from such a group and comparing
è(A)Thank you for your good comments. This is a paired two-group comparison test. Control was the left leg of human A before wearing stocking and intervention was the same-left leg of human A after wearing stocking. Thus, we inserted the following sentence in Materials and Methods.
We asked the participants not to change their lifestyle during a study period.
My second reservation is the lack of reliability testing for the measurements when dealing with such a small sample size. My final reservation is the presented ultrasound images which demonstrate no gained tissue beneath the fascial line. An independent person cannot make a judgement on whether this is the correct position of the fascial line.
è(A) The white line of skin bottom is fascia!! Therefore, the distance between 2 arrows includes the epidermis, dermis and subcutaneous tissue. In the previous our-paper, we described the followings: When we measured the skin thickness, we had the women repeatedly perform anteflexion and retroflexion of the ankle. The skin consists of the epidermis, dermis and subcutaneous layers above the fascia and muscle. Therefore, when the women moved their ankle, the fascia and muscle layers moved as well, while the skin layer did not move allowing us to easily identify the skin layer (Yanagisawa, M.; Koshiyama,M.; Watanabe, Y.; Sato, S.; Sakamoto, S. A quantitative method to measure skin thickness in leg edema in pregnant women using B-scan portable ultrasonography: a comparison between obese and non-obese women. Med Sci Monit. 2019, 25, 1-9)
The paper is written well and is beneficial in nature however the figures are not easily discernable and I would suggest finding a more appropriate way to display your results prior to publicaiton.
è(A)Yes. Thank you for your good advices. We ask the journal office to arrange the Figure size.

Round 2
Reviewer 1 Report
The authors responded to some suggestions while others did not, that are the following questions:
1. Who performed the ultrasound evaluation? For example, a sonographer with what experience or specialization, specialization, an expert ecc...
2. What is the reliability of these measurements? The reliability of the measurements must be calculated.
Reviewer 2 Report
The authors gave the reply to my comments. But I think that the minimal revised manuscript by authors is not improved and still suboptimal for scientific paper.
1. There is no control group in the study design in the revised manuscript. The authors’ comment is not about control group but baseline study.
2. Evaluation of edema by grading system of pitting edema is not need. The authors should evaluate not total thickness of the skin but the subcutaneous tissue thickness.
The echogenic pattern of the tissue also described. Please refer to the article “Suehiro K, et al. Subcutaneous tissue ultrasonography in legs with dependent edema and secondary lymphedema. Ann Vasc Dis. Volume 7, 2014, 21-27.
Reviewer 3 Report
Thank you for your responses. Unfortunately I don't feel that enough changes have been made for me to be able to support publication.
I feel there are still too many potential covariates for the change even though you have added a sentence about lifestyle. My recommendation would be to measure the legs of 24 pregnant women at 37-38 weeks gestation for a comparison rather than having your comparison be the same legs at a different weeks gestation. I am happy to be guided by the editor on this comment however.
I still feel the ultrasound images are suboptimal. You state that the white line is the deep fascia which it does appear to be, however it is impossible for the reader to know this due to the choice of machine settings. Again, I am happy to go with the editors judgement on this.
I would be hesitant to claim reliability when the testing that you reference was done with a different transducer type. Although I agree your method with the linear probe is better, what is the basis for your claim that what you did is more accurate? At the very least I think this discrepancy needs to be addressed within the manuscript.
All parts of Figure 1 need referencing.
Finally, I am very sorry, and I have tried but the results graphs still don't make any sense to me. Figures 2, 3, 5,6. Could you please consider displaying your results in a clearer fashion.
I commend the authors once again on a good manuscript and believe it would be beneficial to reach a point where this is suitable for publication. Thankyou for the opportunity to review this manuscript.